# Microstructure and Nanoindentation Behavior of FeCoNiAlTi High-Entropy Alloy-Reinforced 316L Stainless Steel Composite Fabricated by Selective Laser Melting

**DOI:** 10.3390/ma16052022

**Published:** 2023-02-28

**Authors:** Xinqi Zhang, Dongye Yang, Yandong Jia, Gang Wang

**Affiliations:** 1School of Materials Science and Engineering, Shanghai University of Engineering Science, Shanghai 201620, China; 2Institute of Materials, Shanghai University, Shanghai 200444, China; 3Zhejiang Institute of Advanced Materials, Shanghai University, Jiaxing 314100, China

**Keywords:** selective laser melting, high-entropy alloy, 316L stainless steel, metal matrix composites

## Abstract

Selective laser melting (SLM) is one of the metal additive manufactured technologies with the highest forming precision, which prepares metal components through melting powders layer by layer with a high-energy laser beam. The 316L stainless steel is widely used due to its excellent formability and corrosion resistance. However, its low hardness limits its further application. Therefore, researchers are committed to improving the hardness of stainless steel by adding reinforcement to stainless steel matrix to fabricate composites. Traditional reinforcement comprises rigid ceramic particles, such as carbides and oxides, while the research on high entropy alloys as reinforcement is limited. In this study, characterisation by appropriate methods, inductively coupled plasma, microscopy and nanointendation assay, showed that we successfully prepared the FeCoNiAlTi high entropy alloy (HEA)-reinforced 316L stainless steel composites using SLM. When the reinforcement ratio is 2 wt.%, the composite samples show higher density. The SLM-fabricated 316L stainless steel displays columnar grains and it varies to equiaxed grains in composites reinforced with 2 wt.% FeCoNiAlTi HEA. The grain size decreases drastically, and the percentage of the low angle grain boundary in the composite is much higher than in the 316L stainless steel matrix. The nanohardness of the composite reinforced with 2 wt.% FeCoNiAlTi HEA is twice as high as the 316L stainless steel matrix. This work demonstrates the feasibility of using a high-entropy alloy as potential reinforcement in stainless steel systems.

## 1. Introduction

The 316L stainless steel is an austenitic steel with face-centered cubic structure. The 316L stainless steel has gained extensive attention in commercial manufacture and structural applications on account of its excellent corrosion resistance, high temperature resistance, creep resistance, formability and biocompatibility [1,2,3]. However, the relatively low yield strength and poor wear resistance would increase the risk of failure during actual application [4]. Recently, many routes have been explored to significantly improve the strength of metals and alloys, and a feasible way to improve the strength of 316L stainless steel is to add hard reinforcement to the 316L matrix and fabricate composites [5]. Casting and powder metallurgy are the traditional preparation methods of metal matrix composites (MMCs) [6,7]. However, inhomogeneous composition distribution and coarse-grained microstructure are usually observed in stainless steel composites prepared by conventional methods. Moreover, these traditional fabrication techniques are time-consuming and costly. 

Additive manufacturing, as a new type of fabrication method, is used to produce MMCs, due to the short production cycle and ability to form complex structural parts [8]. Selective laser melting (SLM) is one of additive manufacturing processes, which prepares metal components through melting powders layer by layer with a high energy laser beam. Usually, the strength and hardness of 316L stainless steel fabricated by SLM is higher than 316L stainless steel prepared by traditional methods [9,10,11]. The microstructure is the main factor which affects the hardness of metal materials. Due to the extremely high cooling rate during the SLM process (about 10^3^–10^8^ K/s), materials fabricated by SLM typically exhibit a fine microstructure that increase the strength and hardness [12,13,14]. As is well-known, the process parameters of the SLM process affect the microstructure and mechanical properties remarkably. Hardik et al. [15] reported that the hardness mainly depended on the layer thickness and the hardness of the materials increased with the layer thickness decreased from 50 μm to 30 μm. Tucho et al. [16] found that the hardness of 316L stainless steel manufactured by SLM increased linearly with the increase in energy density in the range of 50–125 J/mm^3^. Parth et al. [17] reported the effects of layer thickness and angle orientation on hardness. Stainless steel with excellent hardness (218.835 BHN) can be obtained at the lower layer thickness and the higher angle orientation. At this time, the layer thickness was 30 μm and the angle orientation was 90°.

Since ancient times, various alloys have played an important role in production. The production of alloys dates back to Bronze Age. The design concept of traditional alloy is based on one element and uses minor other elements to obtain alloys with enhanced properties. For example, steel is one of the most common alloys. Steel is an alloy consisting of iron (the major element) with C, Si, Mn, P, S and minor other elements. In addition to Fe, the content of C plays a major role in the mechanical properties of steel, so it is collectively referred to as iron–carbon alloy. It is the most important and most used metal material in engineering technology. However, people have never stopped developing new materials. HEA employs five or more elements with atomic fractions between 5% and 35%, which has a completely different design concept from conventional alloys [18,19,20]. The HEAs with reasonable design fully combine four effects and exhibit excellent properties, such as high hardness, superb specific strength, corrosion resistance, wear resistance, resistance to high temperature softening and novel functional properties [21,22]. There is no doubt that HEAs have great potential in developing functional and structural materials in the future due to their unique combinations of properties [23]. 

At present, HEAs have aroused the interest of researchers [24,25,26]. However, taking the short history of research on HEAs into consideration, the field is still in its infant stage. It is necessary to actively focus on improving the performance of HEAs through various approaches, such as surface modification technologies, carburizing, boronizing, etc. [21,27,28,29]. Nevertheless, the great potential of using HEAs as a reinforcement cannot be ignored. According to our knowledge, carbides are mostly chosen as reinforcements to strengthen stainless steel [30,31,32]. However, research on using novel materials such as high entropy alloy (HEA) as reinforcements is limited. As previously described, the high hardness and strength can make up for the shortcomings of 316L stainless steel. Moreover, the HEAs are metallic in nature, and metallic reinforcements often offer excellent bonding with the matrix [33,34,35,36].

In this study, the FeCoNiAlTi high-entropy alloy was selected as reinforcement and FeCoNiAlTi high-entropy alloy-reinforced 316L stainless steel metal matrix composites were fabricated by SLM. The effect of laser scanning speed and the ratio of FeCoNiAlTi HEA reinforcement on the microstructure and nanoindentation behavior were investigated systematically. The reasonable mass fraction of high entropy alloy reinforcement was determined according to the porosity analysis results. Additionally, it is thus essential to examine the microstructure and nanohardness. Hence, the aim of this work is to prove the feasibility of the FeCoNiAlTi high-entropy alloy as reinforcement for 316L stainless steel.

## 2. Materials and Methods

In this study, commercial 316L stainless steel powders with a particle size of 15–53 μm were selected as matrix materials. The FeCoNiAlTi HEA powders with particle size of 15–53 μm (Zhongyuan New Material Technology Co., Ltd., Ningbo, China) were added as reinforcement. Figure 1a,b shows the SEM image of HEA powders and the particle size distribution of HEA powders. The chemical compositions of the HEA powders were given in Table 1. The chemical composition of the HEA powder was analyzed by an inductively coupled plasma optical emission spectrometer (ICP-OES)(5110, Agilent, America). In addition, the particle size distribution of HEA powder was determined by a laser particle size analyzer (LS-13, Beckman Coulter, America). The reinforcement ratios of HEA powders were set to be 2 wt.%, 5 wt.% and 10 wt.%, respectively. Both 316L powders and HEA reinforcement powders were mixed according to the ratio in a three-dimensional mixer (SYH-10L, Changzhou, China) to obtain composite powders. The mixing process was ordinary three-dimensional motion mixing, the speed of the mixer was 45 r/min, the environmental condition was room temperature and the mixing time was 2 h. Figure 1c shows the SEM image of the mixed powders (FeCoNiAlTi HEA and 316L powders).

The composites were fabricated in an HBD-100 SLM device (Shanghai Hanbang, China) equipped with a maximum laser power of 200 W, and 60 μm spot diameter. The type of laser is single mode fiber laser. The scanning speeds were 650 mm/s, 800 mm/s and 1000 mm/s, respectively. The selection of parameters was based on the literature and the guidance from skilled machine operators [17]. A stainless-steel plate was chosen as the substrate and the whole fabricating process was carried out under the protective argon atmosphere. The dimensions of bulk samples and the scanning strategy were shown in Figure 2, the red arrows in the picture represent the laser scanning direction. The laser energy density (*E*) used to fabricate composites was calculated as:(1)$$E=\frac{P}{h\cdot v\cdot d}$$
where *P* is laser power, *h* is the layer thickness, *v* is the scanning speed and *d* is hatch distance. In this study, *P*, *h* and *d* were kept constant (180 W, 30 μm and 50 μm, respectively).

An optical microscope (VHX-600, KEYENCE, Japan) was used to observe the pore of the samples. A scanning electron microscope (SEM, MIRA LMS, TESCAN, Czech) was used to record the microstructures of the composites fabricated by SLM. The crystallographic information was collected by backscattered electron diffraction (EBSD, Hikari Plus, EDAX, America). The acceleration voltage was kept as 20 kV with a step size of 0.4 μm. OIM software was used to analyze the EBSD data.

The hardness of the material was examined by nanoindentation. Nanoindentation equipment (Hysitron TI980, Bruker, Germany) was selected and the experimental conditions were as follows: maximum load of 5 mN, loading rate of 1 mN/s, holding time of 2 s. All specimens were inspected at 6 different points and the average data were taken as the final hardness value. For indentation, the test points are selected along the direction parallel to building direction, with 20 μm spacing. The influence of pores in the samples on the determination of hardness is huge. Therefore, the selection of test points should avoid pores.

All specimens need to be prepared according to standard metallographic procedure before the test. In terms of microstructural observation, the samples were mechanically ground with SiC papers and polished with a 1 μm diamond polishing paste to achieve a parallel and smooth plane. Additionally, the samples were etched in a solution of 25 mL HNO3 and 75 mL HCL for 10 s. In this study, all microstructure characterization and hardness testing were performed on surfaces which are parallel to the building direction.

## 3. Results and Discussion

### 3.1. Microstructure Characterization

Figure 3 shows the optical microstructure (OM) of the specimens fabricated by SLM. The relative density of the samples is marked in the corresponding surface morphology. For pure 316L stainless steel, the maximum relative density of 99.6% was obtained with a scanning speed of 650 mm/s, and barely any pores were present. For the HEA/316L composite, the maximum relative density of 99.3% was obtained with the scanning speed of 650 mm/s and reinforcement ratio of 2 wt.%. The porosity shown in OM images usually achieves a high degree of matching with the relative density measurement result [37,38]. Therefore, the analysis of pore distribution is helpful to understand the influence of scanning speed and reinforcement ratio on relative density.

It can be seen that when the amount of reinforcement is 2 wt.% and the scanning speed is 650 mm/s, the composite material achieves the highest density. With the increase in the amount of reinforcement and the scanning speed, the porosity of the sample shows an upward trend. It was reported that the laser absorptivity of 316L stainless steel powders is about 0.6 during the SLM process due to its reflection on light [39]. The addition of reinforcement further reduces the laser absorptivity of the composite powders, resulting in more metallurgical defects. Similar phenomena have been reported by Zhao et al. [40].

At present, compared with the traditional production processes, such as casting and powder metallurgy, one of the obstacles that limits the further development of SLM is the inherent porosity [41]. Additionally, this is also the bottleneck that other additive manufacturing processes are facing. As is well known, the existence of pores deteriorates the performances of material. Litton et al. [42] reported that the defects such as porosity in Ti6Al4V parts fabricated by SLM increase the stress and eventually deteriorate the fatigue performance. Zhao et al. [43] also reported that the presence of pores in SLM processed 18Ni300 maraging steel reduced the plasticity and corrosion resistance. Obviously, the pores in samples will undoubtedly have bad effects on the mechanical and chemical properties of the parts fabricated by SLM. This is also one of the main reasons why many scholars are committed to improving the density of samples prepared by SLM. In the surface morphology shown in Figure 3, when the reinforcement ratio was 2 wt.%, the composite sample had the fewest pores. Therefore, 2 wt.% is considered to be the appropriate reinforcement ratio due to the relative lower porosity in the corresponding specimens. In the following, all the studies of composites are based on the composites reinforced with 2 wt.% HEA.

It is known that scanning speed remains one of the crucial parameters that affects the presence of defects in the SLM processed samples [44]. The metallurgical defects observed in the SLMed composites as a function of varying scanning speeds are shown in Figure 4. The number and size of the pores increase with the increase in the scanning speed due to the Marangoni effect [45,46,47]. The gas generated during the SLM process escapes from the molten pool through the convection process known as the Marangoni convection. However, the increase in laser scan speed decreases the laser power density, weakens the Marangoni effect and leads to gas entrapment in the melt pool [48]. Increased laser scan speed decreases the overall temperature of the melt pool and hence results in a shorter solidification time. Accordingly, the gas becomes trapped as spherical pores, which are also known as metallurgical pores [49]. Additionally, the rapid scanning speed may lead to an unstable melt flow, which can also introduce pores into the solidified components [50]. It can also be observed from Figure 4 that the cracks present with increased scanning speed, and the size increases from 15 to 183 µm when the scanning speed increase from 800 to 1000 mm/s.

Generally, the SLM microstructure is not uniform, as a result of its characteristics, and thus displays anisotropy along its length scale [51,52,53]. The microstructure of different regions within a melt pool are shown in Figure 5. From Figure 5a, the presence of cellular structure or known as microstructure can be observed in both regions I and II (with different sizes however). On the other hand, in region III, the columnar structure is observed because this region is located next to the hatch overlaps [54,55,56]. Consequently, from the center of the melt pool to the boundary, an anisotropy is observed, where fine cellular microstructure transforms into a columnar structure along the boundary. The present results are consistent with the results reported by Kimura et al. [57], where the relatively slow cooling rate observed along the hatch overlaps due to a double melting event that makes the microstructure coarser and leads to a columnar morphology. It can be observed from Figure 5b that a fine cellular structure (~430 nm in diameter) is observed along the core of the melt pool and a relatively coarse cellular structure (~1.69 µm in diameter) is presented in region II, which is the intermediate region between the core and the hatch overlaps. Such variation in the microstructural features may be attributed to the changes in the cooling rate and the decrease in temperature (heat input) with the increase in distance from the center of the molten pool due to the Gaussian distribution of the laser beam [58].

In summary, the change in microstructure within the melt pool can be related to the changes in the G/R ratio (where G corresponds to the temperature gradient and R is the cooling rate) [59,60]. A low value of the G/R ratio contributes to the formation of a cellular structure, and with gradual increase in G/R is observed as the distance from the center of the melt pool increases, which leads to relatively coarser grains (grain growth may be observed) and finally causes cellular to columnar transformation in and near the hatch overlaps.

Figure 6 shows the microstructure and EDS distribution of the SLM-fabricated composites with different scanning. All samples show a cellular structure that is typical for 316L stainless steel prepared by SLM. Such a metastable microstructure is a result of a high cooling rate and unique non-equilibrium conditions that exist during the SLM process [61]. In addition, HEA reinforcements that dispersed in the matrix can be observed in all samples (indicated by yellow arrows). It is supposed that the reinforcement particles are able to promote heterogeneous nucleation, resulting in grain refinement. Meanwhile, the addition of reinforcement leads to a pining effect as well, which is conducive to grain refinement [30]. Additionally, Figure 6d–h shows the EDS distribution of the main HEA elements of the composite sample fabricated at 650 mm/s. It is not hard to find that although HEAs contain more constitutive elements, the elements uniformly distribute in the composite fabricated via SLM. Since the phase formation after the SLM process depends on the homogeneity of these elements. Therefore, the results of EDS imply the uniform phase composition and good metallurgical bonding.

Figure 7 shows the inverse pole figure (IPF), the grain boundary distribution (GBD) and the average grain size and distribution maps of pure 316L stainless steel and its composite fabricated via SLM at 650 mm/s. Obviously, columnar grains are observed in Figure 7a, which is a typical phenomenon of SLM-fabricated samples as a result of epitaxial growth [62]. The reason for the formation of columnar grains is the temperature gradient along the building direction [31]. The microstructure of the 2 wt.% HEA-reinforced composite varies from columnar grains to equiaxed grains, as shown in Figure 7b, which indicates that the change of nucleation mode and growth mode due to the presence of reinforced particles in the matrix. The grains of pure 316L stainless steel and its composites are randomly oriented. The main reason of this random grain orientation patterns is the rotation of scanning direction and the periodic thermal cycle [63,64]. The GBD maps of SLM-fabricated 316L stainless steel and its composites are shown in Figure 7b,e. The orientation difference of adjacent grains of less than 10° is defined as the low angle grain boundary (LAGB), and the orientation difference of adjacent grains higher than 10° is defined as high angle grain boundary (HAGB). Compared with the pure 316L stainless steel, the composite displays higher volume fraction of LAGBs than the 316L stainless steel matrix. The migration of LAGBs is more difficult than the HAGB; therefore, a high LAGB ratio will have a positive effect on structural stability [65,66]. Additionally, a higher percentage of LAGBs can more effectively hinder the movement of dislocations during deformation and eventually lead to an increase in strength [67]. Therefore, a higher percentage of LAGB is helpful to enhance the strength and hardness of stainless steel matrix. Figure 7c,f shows that the average grain size of pure 316L stainless steel and its composite are 15.86 μm, and 0.66 μm, respectively. The grain size of composites is much smaller than that of the matrix material and reaches the nanoscale.

### 3.2. Hardness Testing

Figure 8a shows the displacement-load curve of the SLM-fabricated composites. The value of Young’s modulus (*Er*), hardness (*H*) and contact stiffness (*S*) based on the displacement-load curve is summarized in Table 2. The hardness values obtained from other relevant studies are mentioned in Table 2 for comparison. The comparison of hardness between 316L stainless steel and its composites prepared by SLM is shown in Figure 8b. It can be seen that the hardness of the 316L stainless steel fabricated with scanning speeds of 650, 800 and 1000 mm/s are 2.51 GPa, 2.63 GPa and 2.49 GPa, respectively. Accordingly, the hardnesses of the 2 wt.% HEA-reinforced composites are 5.70 GPa, 5.86 GPa and 5.65 GPa, respectively. The EBSD results have certified that the grain size of the stainless steel matrix is significantly reduced after the addition of the reinforcement. This is considered the main reason for the increase in composite hardness. Figure 8c–e shows nanoindentation traces presented in Figure 8a, respectively. No defects, such as pores and cracks, were observed around the indentation, indicating that the selection of test points was reasonable. The imprints have a standard triangle shape, which prove the dominant plastic deformations of HEA/316L composites during nanoindentation. As shown in Figure 5, samples fabricated by SLM have complex molten pool structures, which lead to different regions with different grain sizes in the same sample. Therefore, the changes in the experimentally determined values are probably due to the complex molten pool structure.

## 4. Conclusions

In this study, FeCoNiAlTi HEA-reinforced 316L stainless steel matrix composites were successfully fabricated by SLM, and the following conclusions can be drawn:(1)Based on the analysis of porosity and metallurgical defects, it can be observed that lower reinforcement ratio and scanning speed contribute to preparing the composites with high density and fewer defects.(2)As a result of epitaxial growth, the SLM-fabricated 316L stainless steel displays columnar grains with a size of about 15.86 μm, whereas the microstructure of the 2 wt.% HEA-reinforced composite varies from columnar grains to equiaxed grains with a size of about 0.66 μm. The percentage of the low angle grain boundary in the composite is much higher than that in the 316L stainless steel matrix. The addition of HEA not only promotes the transformation from columnar crystals to equiaxed crystals but also refines the grains. Additionally, the distribution of HEA elements may imply a uniform phase contribution and good metallurgical bonding.(3)The main contribution of this paper is that the feasibility of using HEA as a reinforcement in fabricated stainless steel composites with high hardness by SLM is proven through hardness test.(4)In the future, the specific strengthening mechanism and the effect of microstructure on hardness need to be studied. In addition, other tests, such as tensile, wear and corrosion, could be carried out to make the material design concept more meaningful.

## Figures and Tables

**Figure 1 materials-16-02022-f001:**
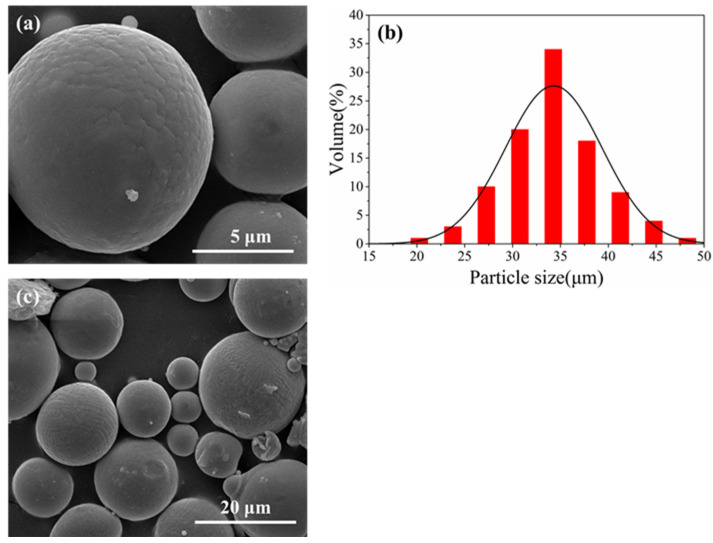
(**a**) SEM image of FeCoNiAlTi HEA powders, (**b**) the particle size distribution of FeCoNiAlTi HEA powders, (**c**) the mixed powders (FeCoNiAlTi HEA and 316L powders).

**Figure 2 materials-16-02022-f002:**
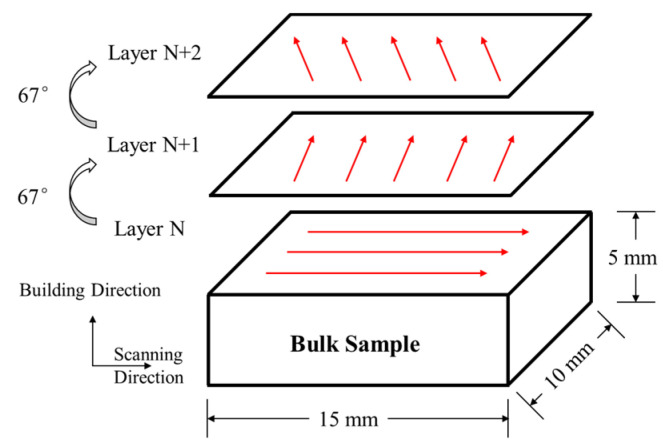
Schematic diagram of additive manufactured preparation process.

**Figure 3 materials-16-02022-f003:**
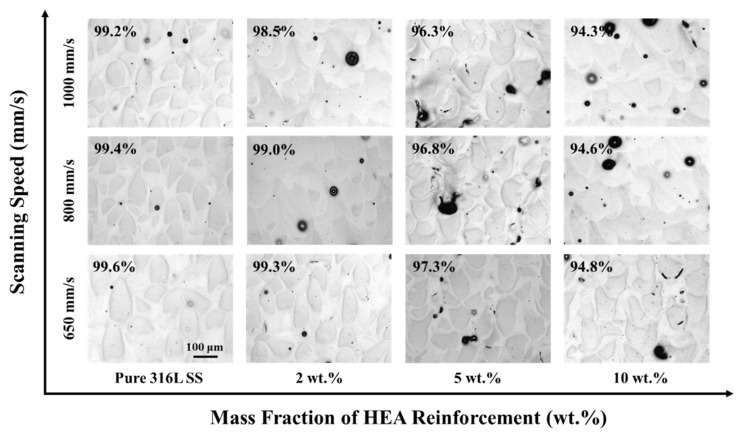
The distribution of the porosity with different reinforcement ratio and scanning speed.

**Figure 4 materials-16-02022-f004:**
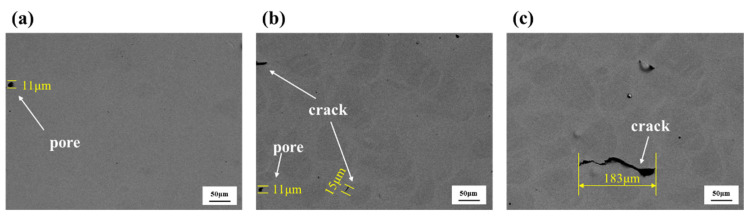
SEM images of defects in SLM-fabricated composites with three different scanning speeds: (**a**) 650 mm/s, (**b**) 800 mm/s, (**c**) 1000 mm/s.

**Figure 5 materials-16-02022-f005:**
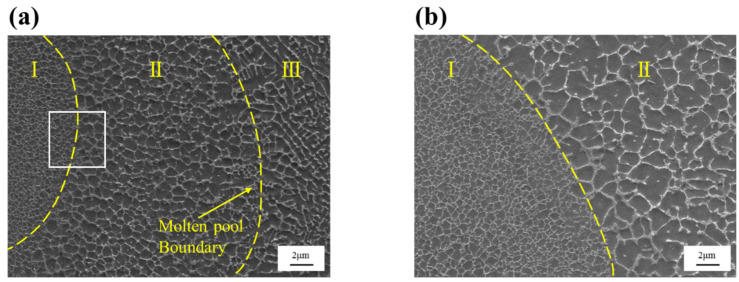
Microstructure of SLM-fabricated composites: (**a**) the three different regions (I, II and III) within a melt pool, and (**b**) the boundary between regions I and II.

**Figure 6 materials-16-02022-f006:**
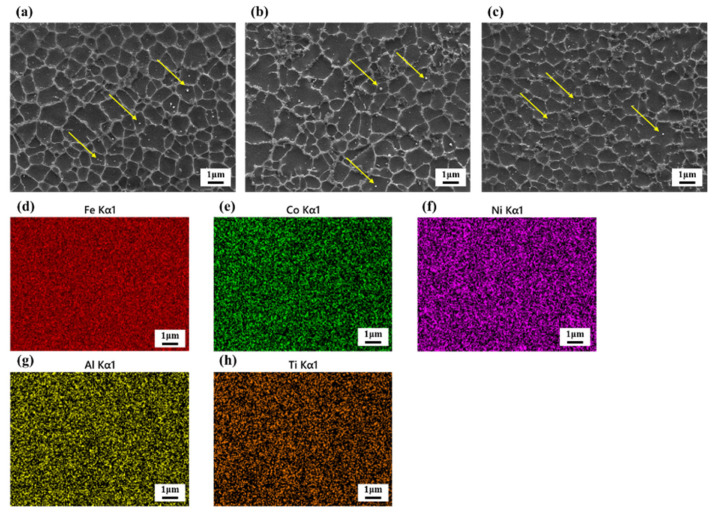
SEM images and the EDS of the SLM-fabricated composites with different scanning speeds: (**a**) 650 mm/s, (**b**) 800 mm/s and (**c**) 1000 mm/s. (**d**–**h**) The EDS distribution of the SLM-fabricated composites at 650 mm/s.

**Figure 7 materials-16-02022-f007:**
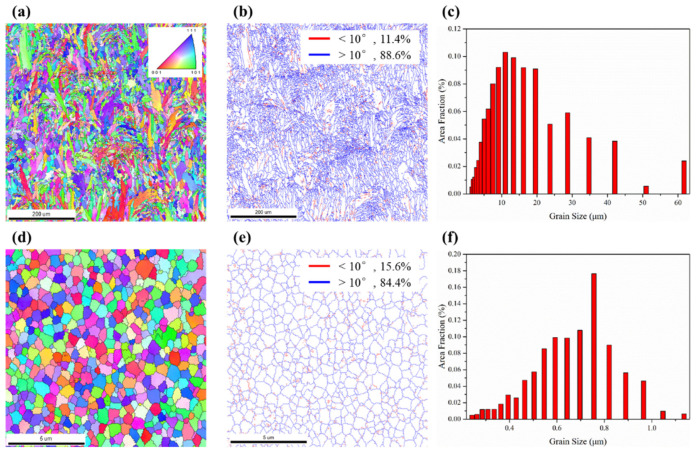
EBSD IPF maps, GBD maps and the average grain size and distribution maps of SLM-fabricated samples: (**a**–**c**) 316L stainless steel and (**d**–**f**) 2 wt.% HEA-reinforced composite.

**Figure 8 materials-16-02022-f008:**
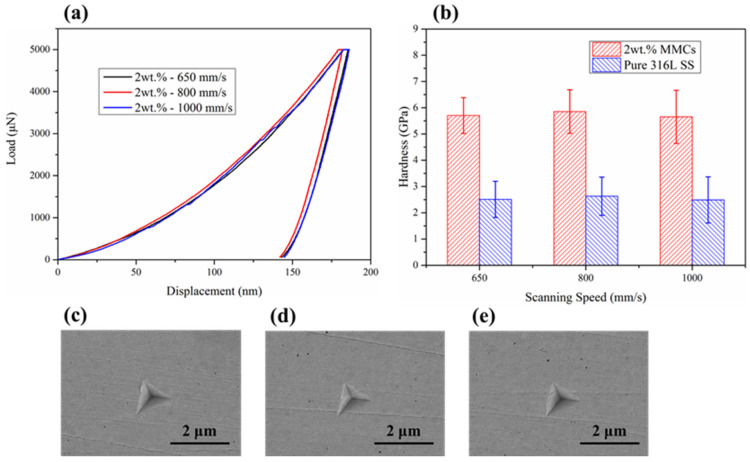
(**a**) the displacement–load curves of the SLM-fabricated composites; (**b**) the comparison of hardness between 316L stainless steel and its composite; and nanoindentation traces of (**c**) 650 mm/s, (**d**) 800 mm/s and (**e**) 1000 mm/s.

**Table 1 materials-16-02022-t001:** Chemical composition of FeCoNiAlTi HEA powders.

Elements	Fe	Co	Ni	Al	Ti
wt.%	28.70	31.90	29.60	4.20	5.60

**Table 2 materials-16-02022-t002:** Young’s modulus (*E_r_*), hardness (*H*) and contact stiffness (*S*) of 2 wt.% HEA-reinforced 316L composites, and the hardness values obtained from other studies.

Molding Methods	Materials	E_r_ (GPa)	H (GPa)	S (μN/nm)	Source
SLM-scanning speed = 650 mm/s	2 wt.% FeCoNiAlTi/316L Composites	157.31	5.70	166.21	This work
SLM-scanning speed = 800 mm/s		169.52	5.86	176.78	
SLM-scanning speed = 1000 mm/s		154.50	5.65	163.95	
SLM	CoCrFeMnNi	/	2.84 ± 0.13	/	[68]
Powder Plasma Arc Additive Manufacturing	CoCrFeNi	/	4.578	/	[69]
	CoCrFeNiTa_0.4_	/	5.672	/	
	CoCrFeNiAl_0.4_	/	4.674	/	
	CoCrFeNiNb_0.4_	/	5.617	/	
Laser Cladding	AlCoCrFeNi	/	6.14 ± 2.06	/	[70]
Laser Powder Bed Fusion	316L	/	2.83 ± 0.091	/	[71]
	1 wt.% CNT(carbon nanotube)/316L Composites	/	4.27 ± 0.21	/	
	5 wt.% CNT/316L Composites	/	7.41 ± 0.55	/	
Spark Plasma Sintering	1 wt.% Gd/316L Composites	/	7.91	/	[72]
	3 wt.% Gd/316L Composites	/	7.34	/	
	5 wt.% Gd/316L Composites	/	5.84	/	

## Data Availability

The data in this work are available from the corresponding authors on reasonable request.

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
