# Peer review of "Microstructure and Nanoindentation Behavior of FeCoNiAlTi High-Entropy Alloy-Reinforced 316L Stainless Steel Composite Fabricated by Selective Laser Melting"

_materials, 2023, doi:10.3390/ma16052022_

Round 1

Reviewer 1 Report

SUGGESTIONS TO AUTHORS

The major revision is needed for this paper. Please find the following suggestions:

- The abstract must be rewritten.

- The reference number should be used in number.

- The introduction is poor. It must be expanded to cite the newest works in this field and find out the research gap.

- What does “…………..” in Eq. (1) mean?

- The experimental set up was not provided, so please add and describe in detail.

- What is the basis for choosing the scanning speeds of 650 mm/s, 800 mm/s, and 1000 mm/s.

- Please add the manufacturer name and specification for HBD-100 SLM device.

- The conclusion section must be improved, and the future work should be added. What are the main contributions?

- The paper still contains many grammatical errors. Please revise carefully.

Author Response

Dear reviewer,

Manuscript ID: materials-2189789

Title: Microstructure and nanoindentation behavior of FeCoNiAlTi high entropy alloy reinforced 316L stainless steel composite fabricated by selective laser melting

We are grateful for the thoughtful comments. The valuable and insightful comments help us further improve our manuscript. We have studied comments carefully and have made correction which we hope meet with approval. Following our careful revision, we believe that it will match this high-level journal.

Comment 1: The abstract must be rewritten.

Response: We have rewritten the abstract by adding the methods used in this work and highlighting the importance of the problem studied and the novelty.

Comment 2: The reference number should be used in number.

Response: We are sorry about our mistake and have changed it.

Comment 3: The introduction is poor. It must be expanded to cite the newest works in this field and find out the research gap.

Response: Many thanks for the comments and suggestions from the reviewer. We have changed it.

Comment 4: What does “…………..” in Eq. (1) mean?

Response: We are sorry about our mistake and have deleted it.

Comment 5: The experimental set up was not provided, so please add and describe in detail.

Response: We add the detail about the SLM device.

Comment 6: What is the basis for choosing the scanning speeds of 650 mm/s, 800 mm/s, and 1000 mm/s.

Response: The selection of parameters based on literature and guidance from skilled operators.

Comment 7: Please add the manufacturer name and specification for HBD-100 SLM device.

Response: The manufacturer name and the type of laser has been added.

Comment 8: The conclusion section must be improved, and the future work should be added. What are the main contributions?

Response: Many thanks for the comments and suggestions from the reviewer. We have added the future work and emphasized the main contribution.

Comment 9: The paper still contains many grammatical errors. Please revise carefully.

Response: Many thanks for the comments and suggestions from the reviewer. We have changed it.

Reviewer 2 Report

The authors investigated the effect of FeCoNiAlTi high entropy alloy reinforcement to improve the mechanical properties of AISI 316 stainless steel produced by selective laser melting process. The article contains novelty. However, the fact that only the hardness and elasticity modules were examined in the article limits the article. If additional tests such as wear, tensile, bending etc. could be done, the article would be much more meaningful. Even if it is not possible to carry out these tests, the article can be published by making the following revisions, since the article contains novelty of HEA alloys and thus contributes to the literature.

Since the novelty part of the article is an addition to HEA, please give a paragraph about the importance of HEA in the introduction part. You can use the following articles.

https://doi.org/10.1016/j.jallcom.2021.161222

https://doi.org/10.1016/j.mattod.2015.11.026

Figure 3 should be replaced with etched microstructure pictures whenever possible.

Compare the hardness values in Table 2 with the HEAs deposited with the laser clad and other mothods.

https://doi.org/10.1016/j.surfcoat.2022.128830.

https://doi.org/10.1016/j.addma.2019.06.012

https://doi.org/10.3390/nano11030721

Please, give 3 of the nanoindentation traces presented in Figure 8.

Author Response

Dear reviewer,

Manuscript ID: materials-2189789

Title: Microstructure and nanoindentation behavior of FeCoNiAlTi high entropy alloy reinforced 316L stainless steel composite fabricated by selective laser melting

We are grateful for the thoughtful comments. The valuable and insightful comments help us further improve our manuscript. We have studied comments carefully and have made correction which we hope meet with approval. Following our careful revision, we believe that it will match this high-level journal.

Comment 1: Since the novelty part of the article is an addition to HEA, please give a paragraph about the importance of HEA in the introduction part. You can use the following articles.

Response: Many thanks for the comments and suggestions from the reviewer. We have changed it.

Comment 2: Figure 3 should be replaced with etched microstructure pictures whenever possible.

Response: We replace the pictures in figure.3.

Comment 3: Compare the hardness values in Table 2 with the HEAs deposited with the laser clad and other methods.

Response: Many thanks for the comments and suggestions from the reviewer. We have cited the literature and supplemented the relevant data in Table 2.

Comment 4: Please, give 3 of the nanoindentation traces presented in Figure 8.

Response: We give nanoindentation traces in figure 8.

Reviewer 3 Report

The paper presents results of microstructure and nanoindentation behavior of FeCoNiAlTihigh entropy alloy reinforced 316L stainless steel composite via SLM process. The submitted study is very interesting and shows well elaborated results. In order to publish the manuscript, I would like to ask you for some revisions.

1. The previous work is very low, and isn't sufficient. It is recommended updated this section with new references, and compared them with your work.

2. It is necessary to detail which standard metallographic procedure was used.

3. The numbering of the bibliography must be redone according to the MDPI journal.

4. The explanations on hardness testing should be supplemented by the indication of the test positions (e.g. graphically). This improves the understanding of the values determined.

5. Statistical analyses for experimental data would give scientific consistency of the work.

6. When evaluating the results of the hardness test, an explanation for the changes would be useful (point 3.2). What exactly in the microstructure change leads to the changes in the experimentally determined values?

7. The hardness results should be compared with other results obtained in other studies.

8. The conclusions must be more detailed. They are short and do not include all the results obtained.

Author Response

Dear reviewer,

Manuscript ID: materials-2189789

Title: Microstructure and nanoindentation behavior of FeCoNiAlTi high entropy alloy reinforced 316L stainless steel composite fabricated by selective laser melting

We are grateful for the thoughtful comments. The valuable and insightful comments help us further improve our manuscript. We have studied comments carefully and have made correction which we hope meet with approval. Following our careful revision, we believe that it will match this high-level journal.

Comment 1: The previous work is very low, and isn't sufficient. It is recommended updated this section with new references, and compared them with your work.

Response: Many thanks for the comments and suggestions from the reviewer. We have changed it and added the new references.

Comment 2: It is necessary to detail which standard metallographic procedure was used.

Response: Many thanks for the comments and suggestions from the reviewer. We have added it.

Comment 3: The numbering of the bibliography must be redone according to the MDPI journal.

Response: We are sorry about our mistake and have changed it.

Comment 4: The explanations on hardness testing should be supplemented by the indication of the test positions (e.g. graphically). This improves the understanding of the values determined.

Response: We describe the test method and give nanoindentation traces in figure 8.

Comment 5: Statistical analyses for experimental data would give scientific consistency of the work.

Response: Many thanks for the important and instructive comments and suggestions from the reviewer. Detailed statistical analyses relevant with this study will be carried out in our next work.

Comment 6: When evaluating the results of the hardness test, an explanation for the changes would be useful (point 3.2). What exactly in the microstructure change leads to the changes in the experimentally determined values?

Response: We have added the description of the relationship between microstructure and hardness.

Comment 7: The hardness results should be compared with other results obtained in other studies.

Response: Many thanks for the comments and suggestions from the reviewer. We have added it in Table 2.

Comment 8: The conclusions must be more detailed. They are short and do not include all the results obtained.

Response: Many thanks for the comments and suggestions from the reviewer. We have changed it.

Reviewer 4 Report

Dear authors,

The work done here attracts attention and is very interesting. However, there are certain issues and therefore the work cannot be accepted for publication in this form. My comments, including major and minor concerns, are given below:

1.      Please improve Abstract by adding the methods used in this work and highlighting the importance of the problem studied and the novelty.
2. Materials and methods: Please provide information on the properties of the starting powders, such as purity, morphology, size of HEA particles, particle size distribution, etc. XRD analyses of the starting powders and mixture would also be useful.
3. Materials and methods: The preparation of the powder mixture is not well enough described. Therefore, please add milling conditions, such as: BPR, atmosphere, temperature, etc.
4. How was the chemical composition of the HEA powders given in Table 1 determined? Was it measured in this work (describe a method) or provided by the manufacturer?
5. Since HEAs contain more constitutive elements, phase formation after the SLM process depends on the homogeneity of these elements. Therefore, the EDS distribution of the main HEA elements should be shown.

6. P2 L77: Please specify the type of laser.
7. How many samples of composites were produced?
8. P3 L88: Please describe how the samples were prepared for microstructural analysis and hardness measurements, i.e. how did you achieve a parallel plane.
9. It is not entirely clear which microscope was used to obtain Figure 3. Please define.
10. Density of the composites should be determined and correlated with the microstructure and mechanical properties.
11. Please contrast the results of the hardness measurements with the results of similar studies.
12. Figure 3 shows that pores are present. However, their influence on the mechanical properties is neglected. Furthermore, it is discussed as that there are no pores, which is not acceptable.
13. The references should be numbered according to the instructions for authors of the journal Materials.

14. Extensive proofread by a native English speaker is highly recommended.

Best regards

Author Response

Dear reviewer,

Manuscript ID: materials-2189789

Title: Microstructure and nanoindentation behavior of FeCoNiAlTi high entropy alloy reinforced 316L stainless steel composite fabricated by selective laser melting

We are grateful for the thoughtful comments. The valuable and insightful comments help us further improve our manuscript. We have studied comments carefully and have made correction which we hope meet with approval. Following our careful revision, we believe that it will match this high-level journal.

Comment 1: Please improve Abstract by adding the methods used in this work and highlighting the importance of the problem studied and the novelty.

Response: Many thanks for the comments and suggestions from the reviewer. We have changed it.

Comment 2: Materials and methods: Please provide information on the properties of the starting powders, such as purity, morphology, size of HEA particles, particle size distribution, etc. XRD analyses of the starting powders and mixture would also be useful.

Response: Many thanks for the comments and suggestions from the reviewer. We have changed it.

Comment 3: Materials and methods: The preparation of the powder mixture is not well enough described. Therefore, please add milling conditions, such as: BPR, atmosphere, temperature, etc.

Response: We have added details of the preparation of the powder mixture.

Comment 4: How was the chemical composition of the HEA powders given in Table 1 determined? Was it measured in this work (describe a method) or provided by the manufacturer?

Response: We add the method of measuring the chemical composition of the HEA powders.

Comment 5: Since HEAs contain more constitutive elements, phase formation after the SLM process depends on the homogeneity of these elements. Therefore, the EDS distribution of the main HEA elements should be shown.

Response: We have provided the EDS distribution of the main HEA elements in figure.6.

Comment 6: P2 L77: Please specify the type of laser.

Response: We have added the type of laser.

Comment 7: How many samples of composites were produced?

Response: We have prepared 3 pure 316L stainless steel samples and 9 composites samples.

Comment 8: P3 L88: Please describe how the samples were prepared for microstructural analysis and hardness measurements, i.e. how did you achieve a parallel plane.

Response: We add the details of the preparation of sample for microstructural observation and the hardness test.

Comment 9: It is not entirely clear which microscope was used to obtain Figure 3. Please define.

Response: We have added the type of microscope in “Materials and Methods”.

Comment 10: Density of the composites should be determined and correlated with the microstructure and mechanical properties.

Response: We measure the relative density of the composites.

Comment 11: Please contrast the results of the hardness measurements with the results of similar studies.

Response: The hardness values of other studies have been added in Table 2 for comparison, including different HEAs deposited in other methods and other 316L stainless steel composites.

Comment 12: Figure 3 shows that pores are present. However, their influence on the mechanical properties is neglected. Furthermore, it is discussed as that there are no pores, which is not acceptable.

Response: Pores are formed during the preparation of samples. We describe the effect of pores on mechanical properties.

Comment 13: The references should be numbered according to the instructions for authors of the journal Materials.

Response: We are sorry about our mistake and have changed it.

Comment 14: Extensive proofread by a native English speaker is highly recommended.

Response: Many thanks for the comments and suggestions from the reviewer. We have changed it.

Round 2

Reviewer 1 Report

SUGGESTIONS TO AUTHORS

The paper content is improved. The minor revision is needed for this paper. Please find the following suggestions:

- The quality of all the figures in the paper is very low and it must be improved.

- The reference works were not cited in order, so the authors must check them again.

- In conclusion section, the paper still contains some grammatical errors. Please revise carefully. For example, the statement “In this study, FeCoNiAlTi HEA reinforced 316L stainless steel ma-trix composites were successfully fabricated by SLM, the following conclusions can be 315 drawn”.

Reviewer 2 Report

Dear colleagues, I congratulate you. With this revision, your article has been greatly improved compared to the previous version. Therefore, the article can be published in its current form.

Reviewer 3 Report

The paper can be published. The authors modified the paper as required.

Reviewer 4 Report

Dear authors, thank you for accepting the comments and improving the manuscript along these lines. However, there are still some minor things to do:
1. L18-20: I suggest that you expand one sentence, for example: "In this study, characterisation by appropriate methods, inductively coupled plasma, microscopy and nanointendation assay, showed that we successfully..."
2. L20: The sentence seems incomplete: "When the reinforcement ratio is 2 wt.%, the composite samples with higher density"
3. Please enlarge the letters in Fig. 1b.